# HS 3D-SeboSkin Model Enables the Preclinical Exploration of Therapeutic Candidates for Hidradenitis Suppurativa/Acne Inversa

**DOI:** 10.3390/pharmaceutics15020619

**Published:** 2023-02-12

**Authors:** Christos C. Zouboulis, Xiaoxiao Hou, Henriette von Waldthausen, Konstantin C. Zouboulis, Amir M. Hossini

**Affiliations:** 1Departments of Dermatology, Venereology, Allergology, and Immunology, Staedtisches Klinikum Dessau, Brandenburg Medical School Theodor Fontane and Faculty of Health Sciences Brandenburg, 06847 Dessau, Germany; 2European Hidradenitis Suppurativa Foundation e.V., 06847 Dessau, Germany; 3Berlin Brandenburg Center for Regenerative Therapies, Charité–Universitaetsmedizin Berlin, 10178 Berlin, Germany; 4Department of Chemistry and Kavli Institute for Nanoscience Discovery, University of Oxford, Oxford OX1 3QU, UK

**Keywords:** 3D-SeboSkin, hidradenitis suppurativa, SZ95 sebaceous gland cell line, ex vivo, adalimumab, NFκB, proteins, cytokines, autophagy

## Abstract

Despite the rapid development in hidradenitis suppurativa (HS) research, the immediate introduction of potent therapeutic compounds in clinical trials and the lack of definitive outcome measures have led to the discontinuation of potential therapeutic compound studies. HS is a solely human disease, and therefore, the search for preclinical human models has been given priority. The 3D-SeboSkin model, a co-culture of human skin explants with human SZ95 sebocytes as a feeder layer, has been shown to prevent the rapid degeneration of human skin in culture and has been validated for HS preclinical studies. In this work, the HS 3D-SeboSkin model has been employed to characterize cellular and molecular effects of the EMA- and FDA-approved biologic adalimumab. Adalimumab, a tumor necrosis factor-α inhibitor, was shown to target inflammatory cells present in HS lesions, inducing a prominent anti-inflammatory response and contributing to tissue regeneration through a wound healing mechanism. Adalimumab inhibited the lesional tissue expression of TNF-α, IL-3, IL-15, and MCP-3 and downregulated the secretion of IL-1α, IL-5, RANTES, MCP-2, TNF-α, TNF-β, TGF-β, and IFN-γ. In contrast, IL-6 was stimulated. The compound failed to modify abnormal epithelial cell differentiation present in the HS lesions. Patients with Hurley stage II lesions exhibited stronger expression of autophagy proteins in perilesional than in lesional skin. Adalimumab modified the levels of the pro-apoptotic proteins LC3A, LC3B, and p62 in an individual, patient-dependent manner. Finally, adalimumab did not modify the NFκB signal proteins in SZ95 sebocytes and NHK-19 keratinocytes, used to study this specific pathway. The administration of the validated HS 3D-SeboSkin model in ex vivo studies prior to clinical trials could elucidate the individual pathogenetic targets of therapeutic candidates and, therefore, increase the success rates of clinical studies, minimizing HS drug development costs.

## 1. Introduction

Hidradenitis suppurativa/acne inversa (HS) is a significant, inflammatory skin disease of the terminal hair follicle that mostly presents in the apocrine gland-bearing areas of the body [1]. The 2006 inaugural International Conference on HS, in Dessau, Germany [2,3], and the consequent establishment of the Hidradenitis Suppurativa Foundation, Inc. [3], and the European Hidradenitis Foundation e.V. [3] raised awareness for this previously obscure disease. This exposure brought the number of relevant publications from 480 to 3728 in the 16 years following these inaugural events [4]. An increased prevalence, currently estimated at 0.4% worldwide and up to 1% in Europe, can be attributed to more informed diagnoses of HS [5]. EMA and FDA approval of the tumor necrosis factor-α (TNF-α) inhibitor adalimumab (ADA) in 2016 provided the first therapeutic agent against inflammatory lesions of HS [6,7]. Currently, there are more han 150 clinical studies registered, with several phase 3 studies ongoing or completed [8].

Despite this rapid development, a lack of definitive outcome measures has led to failure amongst initial therapeutic candidates [9,10]. Furthermore, adequate preclinical models were until recently lacking. Given that HS is a solely human disease and there are no relevant animal models, human models are required for appropriate preclinical drug testing.

Therefore, attention is currently being devoted to improving outcome measures [11,12] and developing human HS models [13,14]. Such models contain ex vivo systems such as the 3D-SeboSkin model [15,16,17,18]. The 3D-SeboSkin model [15] comprises a co-culture of human skin explants with a human SZ95 sebocyte [19] feeder layer, shown to prevent rapid degenerative events commonly observed during the maintenance of human skin explants in culture. Furthermore, this model retains normal histomorphological characteristics concerning epidermal structure, the basal membrane, and adhesive junctions [20]. The 3D-SeboSkin model has been validated [15] by reproducing ex vivo the differential expression of HS biomarkers found in epidermal and dermal tissue as well as in the appendages of lesional and perilesional skin of HS patients in comparison to healthy skin [21,22,23].

In this work, we treated the HS 3D-SeboSkin model with ADA [7] to study the pattern of ADA efficacy in HS ex vivo. Moreover, we provided evidence that HS 3D-SeboSkin is an appropriate model for ex vivo preclinical exploration of therapeutic candidates for HS.

## 2. Materials and Methods

### 2.1. HS 3D-SeboSkin Model

The 3D-SeboSkin model, developed to maintain human skin in culture [20], has been adapted for ex vivo HS studies [15].

### 2.2. Cells

Human SZ95 sebocytes [19] and normal human keratinocytes (NHK-19) were cultured in Sebomed basal medium (Biochrom, Berlin, Germany) supplemented with 10% fetal bovine serum (BSA), 50 μg/mL gentamycin, 10 ng/mL human epidermal growth factor (EGF), and 1 mM CaCl_2_ at 37 °C and 5% CO_2_ until reaching sub-confluence. Prior to co-cultivation experiments, SZ95 sebocytes were resuspended in serum-free medium (Sebomed basal medium supplemented with 0.1% BSA, 50 μg/mL gentamycin, 10 ng/mL human EGF, 1.5 mM CaCl_2_, 1.5 × 10^−7^ M linoleic acid (LA), and 10^−6^ M retinol). Two hundred thousand SZ95 sebocytes were seeded in 24-well plates and incubated overnight at 37 °C and 5% CO_2_. The following day, the wells were washed twice with Ca^2+^- and Mg^2+^-free phosphate-buffered saline (PBS) and treated with 400 μL serum-free medium for either cell culture or direct contact co-culture experiments.

### 2.3. Skin Specimens

Following written informed consent, full-thickness skin specimens were obtained from 9 Caucasian female and 3 male patients (aged 28–59 years) with HS Hurley stage II–III during surgery. The patients did not present any other inflammatory or endocrinological disorders, including diabetes and thyroiditis, and were not pretreated for HS. Perilesional skin is defined as adjacent to HS lesional skin at a distance of ≥5 cm from the visible inflammation area. The study was approved by the Ethics Committees of the Charité–Universitätsmedizin Berlin (EA4/016/07) and the Brandenburg Medical School Theodor Fontane (E-01-20210222) and was conducted according to the Helsinki Declaration rules.

### 2.4. Co-Culture Experiments

After subcutaneous fat excision, skin specimens were cut into uniform samples with dimensions of 6 mm. HS skin explants were cultured with and without ADA (30 μg/mL; Selleckchem, Munich, Germany) for three days using the HS 3D-SeboSkin model as previously described [15,20].

### 2.5. Tissue and Culture Supernatant Protein Extraction

Full-length proteins were extracted from skin explants maintained in the HS 3D-SeboSkin model for three days and the harvested co-culture supernatants using the Qproteome FFPE Tissue Kit (Qiagen, Hilden, Germany). Protein concentration was determined using the Biorad Protein Assay (Bio-rad, Hercules, CA, USA).

### 2.6. Protein Quantification

For cytokine quantification, proteins were blotted with a Human Cytokine Antibody Array (ab133996; Abcam, Cambridge, UK). The levels of 23 inflammatory cytokines (GCSF, GM-CSF, GRO (αβγ), GRO-α, IFN-γ, IL-1α, IL-2, IL-3, IL-5, IL-6, IL-7, IL-8, IL-10, IL-13, IL-15, MCP-1, MCP-2, MCP-3, MIG, RANTES, TGF-β, TNF-α, TNF-β) were simultaneously measured. For autophagy protein quantification, proteins were blotted with the RayBio C-Series Human Autophagy Array (Raybiotech, Norcross, GA, USA). The levels of 20 human autophagy proteins (ATG12, ATG7, ATG10, ATG13, ATG3, ATG4A, ATG4B, ATG5, Beclin, BNIP3L, DDR2, GABARAP, LC3A, LC3B, LAMP1 (CD107a), p62, NBS1, RHEB, MSK1, and α-synuclein) were simultaneously measured in tissue lysates. For the detection of 45 nuclear factor “κ-light-chain-enhancer” of activated B-cells (NFκB) signal pathway proteins (ASC, BCL-10, CARD6, CD40/TNFRSF5, dAP1/BIRC2, dAP2/BIRC3, FADD/MORT1, Fas/TNFRSF6/CD95, IκBα, IκBε, IKK1/IKKα/CHUK, IKK2/IKKβ, IKKΥ/NEMO, IL-1R1, IL-17RN, IL-18RN, IRAK1, IRF5, IRF8, JNK1/2, JNK2, LTBR/TNFRSF3, Metadherin/AEG-1, MyD88, NFκB1, NFκB2, NGFR/TNFRSF16, P53, P53 (pS46), RelA/p65, RelA/p65 (pS529), c-Rel, SHARPIN, SOCS-6, STAT1p91, STAT1 (pY701), STAT2, STAT2 (pY689), STING/TMEM173, TLR2, TNFRI/TNFRSF1A, TNFRII/TNFRSF1B, TRAF2, TRAIL-R1/TNFRSF10A, TRAIL-R2/TNFRSF10B), blots of human SZ95 sebocytes and human keratinocytes were prepared with the Proteome Profiler Human NFkB Pathway Array (Bio-Techne, Wiesbaden, Germany).

The blots were imaged via a chemiluminescent imager. Blot intensity was quantified using the semi-quantitative software Image J. The ratio of evaluated cytokine to positive control was defined as the expression result.

### 2.7. Statistics

GraphPad 9 was used for data analysis in this study. All results are presented as mean ± standard error of the mean (SEM). The Shapiro–Wilk test was used to examine the distribution of the data. For statistical significance, a t-test was used where the data were normally distributed. Differences of *p* < 0.05 were considered as significant.

## 3. Results

### 3.1. ADA Does Not Modify the NFκB Signal Pathway in Epithelial Cells

To detect the effects of ADA on inflammatory signaling of epithelial cells, human SZ95 sebocytes and NHK-19 cells were pre-incubated for 24 h with the pro-inflammatory fatty acid LA (10^−4^ M) [24] and subsequently for another 24 h with LA, ADA, or LA and ADA (30 µg/mL). After protein extraction, the levels of 45 NFκB signal pathway members were assessed by an antibody array.

Both untreated NHK-19 cells and those cells treated solely with ADA did not express NFκB signal pathway proteins. LA treatment induced low expression of several NFκB signal pathway members, including members of the TNF superfamily pro-apoptotic signal proteins, tumor necrosis factor-related apoptosis-inducing ligand (TRAIL)-R1/TNFRSF10A, TRAIL-R2/TNFRSF10B, and the co-stimulator CD40/ TNFRSF5. The addition of ADA did not modify expression levels previously induced by LA (Figure 1, Appendix A).

Human SZ95 sebocytes exhibited a markedly stronger pro-inflammatory NFκB pathway signal intensity than NHK-19 cells. Baseline expression levels of the pro-inflammatory proteins NFκB1 and CD40/TNFRSF5, the pro-apoptotic protein p53, the anti-apoptotic Fas/TNFRSF6/CD95, and the NFκB activator IκBε were detected in untreated controls. Treatment with LA, ADA, or a combination thereof did not modify the SZ95 protein expression levels (Figure 2, Appendix A).

Interestingly, proteins that have previously been shown to be differentially expressed in HS, such as interleukin-1 receptor type 1 (IL-1R1), IL-1R antagonist (IL-17RN), and IL-18RN [23,25,26], were not regulated by ADA either in SZ95 sebocytes or in NHK-19 cells.

### 3.2. ADA Modifies Tissue Cytokine Expression in the HS 3D-SeboSkin Model

To characterize the expression of inflammatory cytokines in the HS 3D-SeboSkin model, we investigated the expression levels of 23 cytokines in the ADA-treated explants through a cytokine array. Twelve cytokines, namely granulocyte colony-stimulating factor (GCSF), GRO (αβγ), interferon (IFN)-γ, IL-1α, IL-3, IL-8, IL-15, monocyte chemoattractant protein (MCP)-3 (CCL7), RANTES (CCL5), tumor growth factor (TGF)-β, TNF-α, and TNF-β, were detected in HS-involved skin, while granulocyte–macrophage colony-stimulating factor (GM-CSF), GRO-α (CXCL1), IL-2, IL-5, IL-6, IL-7, IL-10, IL-13, MCP-1 (CCL2), MCP-2 (CCL8), and IFN-γ-induced monokine (MIG (CXCL9)) were below detection levels. ADA treatment induced a statistically significant inhibition of TNF-α (*p* < 0.01), IL-3 (*p* < 0.05), IL-15 (*p* < 0.05), and MCP-3 (*p* < 0.05), but not of GCSF, GRO (αβγ), IFN-γ, IL-1α, IL-8, RANTES, TGF-β1, and TNF-β (Figure 3).

### 3.3. ADA Modifies Cytokine Secretion in the HS 3D-SeboSkin Model

All 23 cytokines tested, GCSF, GM-CSF, GRO (αβγ), GRO-α, IFN-γ, IL-1α, IL-2, IL-3, IL-5, IL-6, IL-7, IL-8, IL-10, IL-13, IL-15, MCP-1, MCP-2, MCP-3, MIG, RANTES, TGF-β, TNF-α, and TNF-β, were detected in the culture supernatants of both the control and the ADA-treated skin explants. ADA treatment downregulated the secretion of the cytokines IL-1α (*p* < 0.01), IL-5 (*p* < 0.05), RANTES (*p* < 0.01), MCP-2 (*p* < 0.01), TNF-α (*p* < 0.01), TNF-β (*p* < 0.001), TGF-β (*p* < 0.01), and IFN-γ (*p* < 0.05) and stimulated the secretion of IL-6 (*p* < 0.05) (Figure 4).

### 3.4. ADA Affects Autophagy Procedures in Lesional and Perilesional HS Skin Ex Vivo in the HS 3D-SeboSkin Model

Among the 20 human autophagy proteins studied in lesional HS skin ex vivo using the HS 3D-SeboSkin model, the signals of the microtubule-associated protein light chain 3 (LC3) and p62 exhibited the strongest expression intensity (Figure 5). LC3 is the first mammalian protein described to be specifically associated with autophagosomal membranes and is detected in early autophagosomes [27]. LC3 exists in two isoforms, the cytosolic precursor LC3A and the active form LC3B, both of which were strongly expressed in HS lesional skin. Among the four HS patients studied, patients with Hurley stage II lesions (*n* = 3) exhibited significantly stronger expression in perilesional than in lesional skin, while the patient with Hurley III lesions exhibited stronger apoptotic signals in the lesional than in the perilesional skin.

Overall, ADA modified the levels of the pro-apoptotic proteins LC3A, LC3B, and p62 in an individual, patient-dependent manner (Figure 5, Appendix A).

## 4. Discussion

Current studies of differential gene and protein expression have identified several candidates as targets for HS treatment [23,28,29]. Alas, published therapeutic trials provide limited information concerning the molecular mechanism of these compounds. These trials have demonstrated that ADA treatment induced a downregulation of the anti-apoptotic protein BCL2 in HS lesional skin [29]. Furthermore, a marked reduction in the B cell compartment with a significant decrease in CXCL13 and B cell activating factor (BAFF) expression levels was observed under ADA treatment [30]. Downregulation of matrix metalloproteinase (MMP)-1 and MMP-9 and upregulation of MMP-13 and tissue inhibitor of metalloproteinases (TIMP)-2 levels were detected in the circulation of HS patients treated with ADA [31]. HS patients treated with brodalumab exhibited a reduction in lipocalin (LCN)2 in the skin and IL-17A in serum levels [32]. No changes in inflammatory marker levels were observed in the lesional skin of HS patients receiving apremilast when compared with patients under placebo [28].

The introduction of human ex vivo and in vitro models has provided valuable information concerning the differential expression of genes and proteins under treatment conditions. These data support the understanding of the molecular mechanism of HS therapy candidates. The protein production of pro-inflammatory cytokines TNF-α, IFN-γ, IL-1β, IL-6, and IL-17A was significantly inhibited by ADA, infliximab, ustekinumab, prednisolone, and rituximab, but not by secukinumab [16]. IL-17A, calgranulin C (S100A12), and the activation marker human leukocyte antigen (HLA)-DR were significantly elevated in HS lesional skin and showed a decrease in expression levels when treated with apremilast in an ex vivo HS model [28]. Lenalidomide and the bromodomain and extraterminal inhibitor CPI-0610 reduced ex vivo skin levels of the Th1/Th2/Th9/Treg-associated cytokines IL-2, IL-4, IL-5, IL-9, and IL-21 and of the inflammatory cytokines/chemokines IL-8, IFN-α, macrophage inflammatory protein (MIP)-1α (CCL3), stromal cell-derived factor (SDF)-1α (CXCL12), MCP-1, and GRO-α and supernatant levels of the growth factors platelet-derived growth factor (PDGF)-1, vascular endothelial growth factor (VEGF)-D, and stem cell factor (SCF) and of the inflammatory cytokines/chemokines MIP-1α, IL-1RN, IL-6, and IFN-α [18]. Moreover, a dose-dependent decrease in B cell activation, as measured by a reduction in the proliferation marker Ki67 and the activation marker HLA-DR, was detected under ADA treatment of isolated cell cultures originating from HS lesional skin [30]. Wound healing profile changes, including the inhibition of the MMP pathway, were observed in human macrophages in vitro under ADA treatment but not under treatment with etanercept or certolizumab-pegol [31]. In our study, the HS 3D-SeboSkin model [15,20] has revealed information regarding ADA activity at the tissue and molecular levels. Additionally, this work has demonstrated the importance of similar preclinical studies of HS therapy candidates prior to their introduction in clinical trials.

The fact that ADA reduced ex vivo tissue levels of IFN-γ, IL-5, IL-15, MCP-2, MCP-3, RANTES, and TGF-β but did not modify the canonical NFκB signal pathway [33] in human skin epithelial cells (NHK-19 keratinocytes and SZ95 sebocytes) confirms prior literature concluding that anti-TNF-α therapy targets professional inflammatory cells and markedly attenuates B cell activation but with minimal effect on other inflammatory pathways and cell types [30]. These data could explain the marked decrease in inflammatory nodule and abscess count under ADA treatment [34], and the delayed response or non-response of draining tunnels [35,36].

The selective targeting of professional inflammatory cells by ADA is also supported by the following observations: strong expression of TNF-α and TNF-β in conjunction with ADA-induced downregulation of tissue levels, expression of the pro-apoptotic TRAIL-R1/TNFRSF10A and TRAIL-R2/TNFRSF10B, and inhibition of the pro-inflammatory, cell proliferation-stimulating cytokine IL-3.

In the HS 3D-SeboSkin model, ADA treatment reduced the secretion of the Th17 polarizing cytokines TGF-β1 and IL-1α but induced the secretion of IL-6. ADA did not modify the expression levels of either Th17 cytokines (e.g., IL-17, IL-22) or Th17 downstream effector mediators (e.g., IL-8, CCL-20). This may explain the effects seen under IL-6 secretion, as IL-6 is produced by monocytes, fibroblasts, endothelial cells, adipocytes, and normal human keratinocytes under the influence of IL-17F [37]. IL-6 inhibits the proliferation of regulatory T lymphocytes and activates Th17 cells, thus maintaining inflammation. IL-6 also stimulates the influx of T lymphocytes to the epidermis; moreover, it participates in the process of growth and differentiation of keratinocytes [38]. The high HS tissue levels of IL-6 have been shown to be significantly downregulated with the inclusion of SZ95 sebocytes in the 3D-SeboSkin model [20]. Although IL-6 serum levels were also found to increase in obese rats treated with ADA [39], ADA has been shown to reduce IL-6 circulating levels in long-term treatment studies of patients with HS and psoriasis [40,41].

Regenerative tissues are characterized by autophagy, particularly in the skin adnexa [42,43,44]. In our study, HS lesional skin expressed high levels of autophagosomal proteins, indicating strong regenerative tissue activity. Interestingly, the skin of patients with Hurley stage II, i.e., single scarring lesions separated by healthy tissue, exhibited significantly stronger expression of autophagosomal proteins in perilesional than in lesional skin. The skin of the patient with Hurley stage III, i.e., confluent scarring lesions, exhibited stronger apoptotic signals in the lesional skin. ADA seems to be involved in the regulation of autophagosomal proteins and induction of a wound healing profile, which is in accordance with previous in vivo and in vitro data [31,45].

## 5. Conclusions

The use of ex vivo studies prior to clinical trials for HS, such as the validated HS 3D-SeboSkin [15,20], in combination with existing molecular data [23], could increase the success of therapeutic candidates and minimize the required costs of overall drug development. Such preclinical studies could define the best phenotype for selective clinical response, as demonstrated in the present work. Moreover, the response profile of the registered biologic ADA has been characterized in the present study, namely the targeting of professional inflammatory cells involved in HS lesions and tissue regeneration through a wound healing profile.

## Figures and Tables

**Figure 1 pharmaceutics-15-00619-f001:**
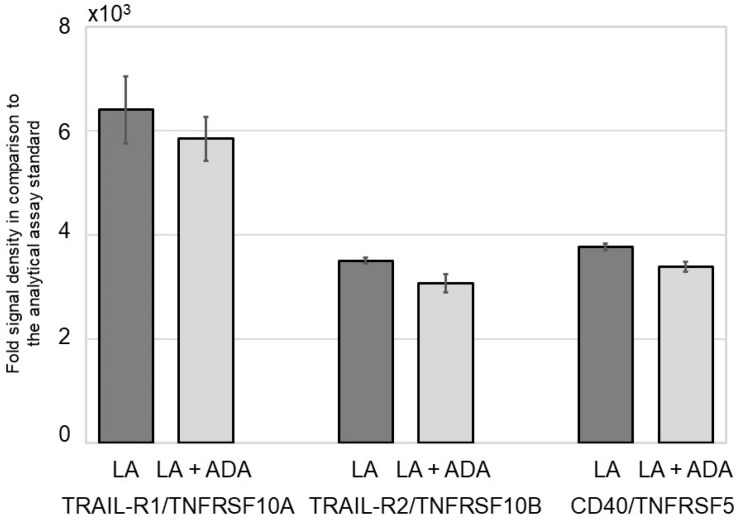
Evaluation of the NFκB signal pathway members TRAIL-R1/TNFRSF10A, TRAIL-R2/TNFRSF10B, and CD40/TNFRSF5 in normal human NHK-19 keratinocytes after 24 h pre-incubation with linoleic acid (LA, 10^−4^ M) and subsequent treatment with LA or LA and adalimumab (ADA, 30 µg/mL) over 3 days. Comparative quantification of the obtained data against the analytical assay standard.

**Figure 2 pharmaceutics-15-00619-f002:**
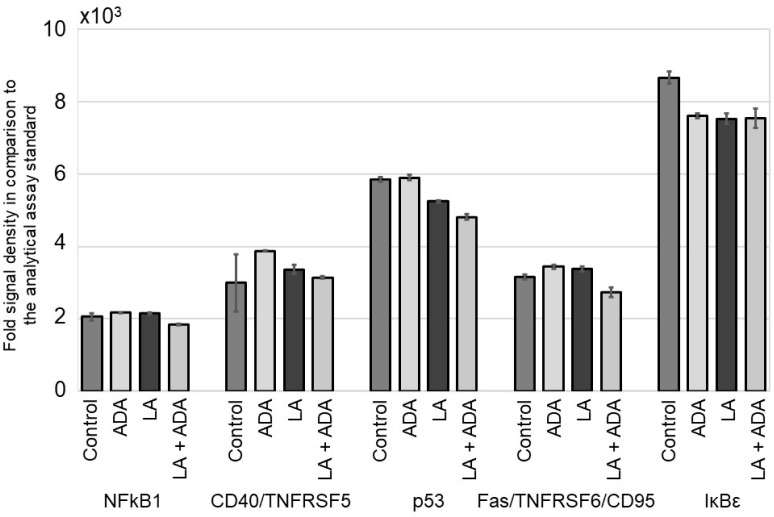
Evaluation of the NFκB signal pathway members NFkB1, CD40/ TNFRSF5, p53, Fas/TNFRSF6/CD95, and IκBε in human SZ95 sebocytes after 24 h pre-incubation with linoleic acid (LA, 10^−4^ M) and subsequent treatment with LA, adalimumab (ADA, 30 µg/mL), or LA and ADA over 3 days. Comparative quantification of the obtained data against the analytical assay standard.

**Figure 3 pharmaceutics-15-00619-f003:**
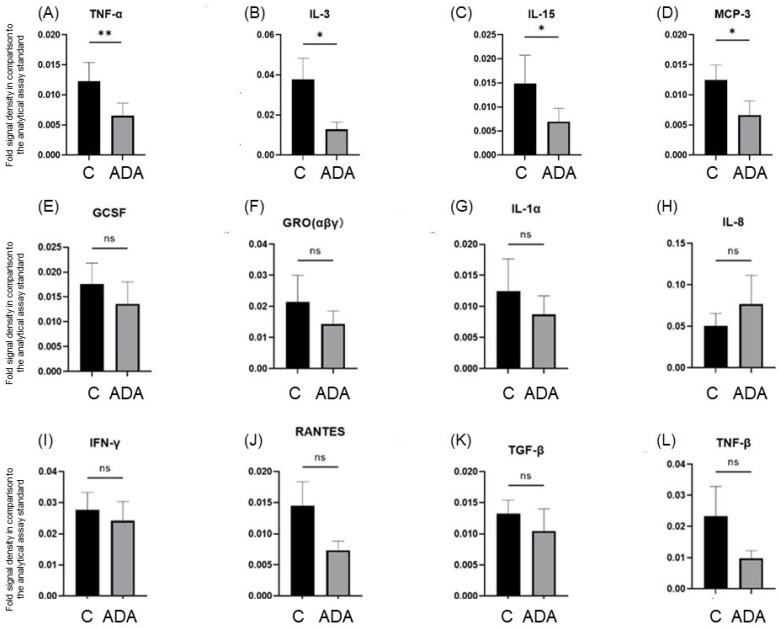
Comparative expression of levels of TNF-α (**A**), IL-3 (**B**), IL-15 (**C**), MCP-3 (**D**), GCSF (**E**), GRO (αβγ) (**F**), IL-1α (**G**), IL-8 (**H**), IFN-γ (**I**), RANTES (**J**), TGF-β (**K**), and TNF-β (**L**) in HS lesional skin of the control group (C) and the adalimumab-treated group (ADA) maintained for 3 days in the HS 3D-SeboSkin model. Comparisons were performed against the analytical assay standard. ** = *p* < 0.01, * = *p* < 0.05, ns = not significant.

**Figure 4 pharmaceutics-15-00619-f004:**
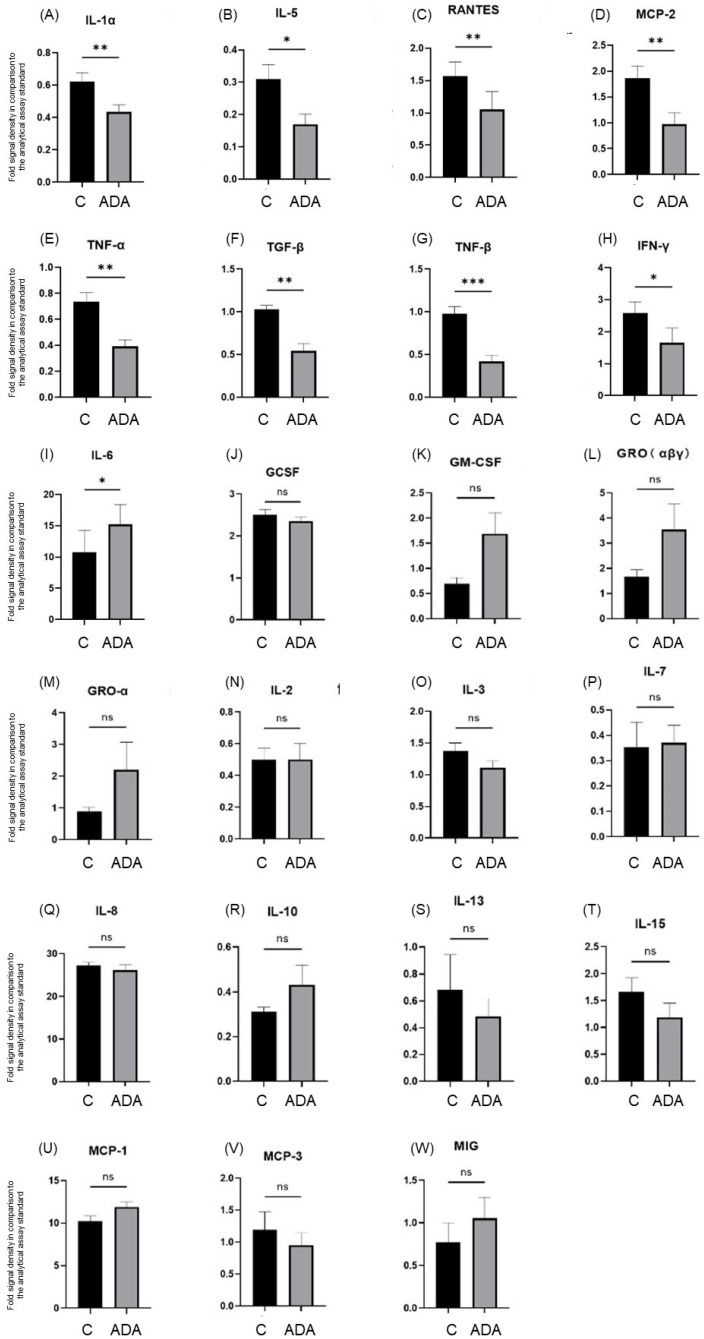
Comparative expression of the levels of IL-1α (**A**), IL-5 (**B**), RANTES (**C**), MCP-2 (**D**), TNF-α (**E**), TGF-β (**F**), TNF-β (**G**), IFN-γ (**H**), IL-6 (**I**), GCSF (**J**), GM-CSF (**K**), GRO (αβγ) (**L**), GRO-α (**M**), IL-2 (**N**), IL-3 (**O**), IL-7 (**P**), IL-8 (**Q**), IL-10 (**R**), IL-13 (**S**), IL-15 (**T**), MCP-1 (**U**), MCP-3 (**V**), and MIG (**W**) in culture supernatants of HS lesional skin of the control group (C) and the adalimumab-treated group (ADA) maintained for 3 days in the HS 3D-SeboSkin model. Comparisons were performed against the analytical assay standard. *** = *p* < 0.001, ** = *p* < 0.01, * = *p* < 0.05, ns = not significant.

**Figure 5 pharmaceutics-15-00619-f005:**
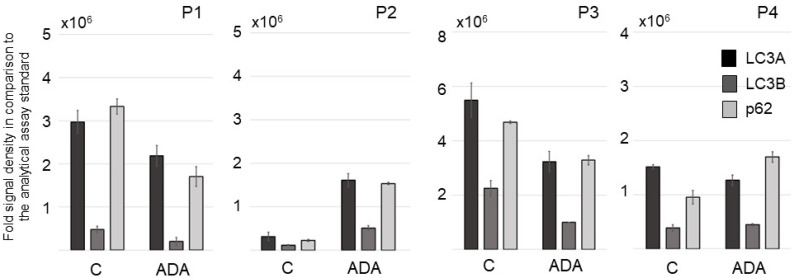
Comparative expression of the levels of the pro-apoptotic proteins LC3A, LC3B, and p62 in three patients with HS Hurley stage II (P1-P3) and one patient with HS Hurley III (P4) in HS lesional skin of the control group (C) and the adalimumab-treated group (ADA) maintained for 3 days in the HS 3D-SeboSkin model. Comparisons were performed against the analytical assay standards.

## Data Availability

The data presented in this study are available on request from the corresponding author.

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
