# Peer review of "HS 3D-SeboSkin Model Enables the Preclinical Exploration of Therapeutic Candidates for Hidradenitis Suppurativa/Acne Inversa"

_pharmaceutics, 2023, doi:10.3390/pharmaceutics15020619_

Round 1

Reviewer 1 Report

This is a valuable body of research that highlights the relevance of 3D-SeboSkin model in exploring molecular/cellular effects of adalimubab therapy. The manuscript is scientifically sound, however, I have a few minor suggestions for the authors. The abstract should be revised to include more information on the key findings of the manuscript. Secondly, the gel blot images in figure 1, 2, 5 are not that critical to the reader in the interpretation of the results and can be shifted to supplementary text.  The y-axis on figure 1 C, 2E, 3, 4, 5 should be labelled.

Author Response

This is a valuable body of research that highlights the relevance of 3D-SeboSkin model in exploring molecular/cellular effects of adalimubab therapy.

On behalf of the authors, I thank the reviewer for his evaluation.

The manuscript is scientifically sound, however, I have a few minor suggestions for the authors. The abstract should be revised to include more information on the key findings of the manuscript.

The abstract has been revised and more information was added.

Secondly, the gel blot images in figure 1, 2, 5 are not that critical to the reader in the interpretation of the results and can be shifted to supplementary text.  

The gel blot images wer shifted to a supplementary file.

The y-axis on figure 1 C, 2E, 3, 4, 5 should be labelled.

The y-axis on new figures 1, 2, 3, 4, 5 were labelled.

Reviewer 2 Report

It is an original and well written manuscript, undoubtedly an interesting topic   on the use of ex vivo studies using HS 3D seboskin model treated with adalimumab prior to clinical trials for HS, that could increase the success of therapeutic candidates and minimize required costs of overall drug development. This manuscript focuses on a crucial aspect of HS study; in fact new appropriate human models are required for preclinical drug testing in therapeutic candidates for HS. 

Author Response

It is an original and well written manuscript, undoubtedly an interesting topic   on the use of ex vivo studies using HS 3D seboskin model treated with adalimumab prior to clinical trials for HS, that could increase the success of therapeutic candidates and minimize required costs of overall drug development. This manuscript focuses on a crucial aspect of HS study; in fact new appropriate human models are required for preclinical drug testing in therapeutic candidates for HS.   

We thank the reviewer for his encouraging comments.

Reviewer 3 Report

Dear Authors, 

I read with interest your paper. It is a well written article including interesting information about a preclinical model for HS. Some points need to be clarified: 

1) What type of surgical procedure was performed in UNTREATED HS patients? It is not common that patients without proper medical treatment to undergo surgery. 

2) Moreover, how perilesional skin was obtained at a distance higher than 5cm? Was it a biopsy or part of the excised surgical piece?

3) The specific area of the body from which the skin samples were taken should be included in the article (armpits, groin, other...).

4) References are not well written. Please rewrite them.

Author Response

I read with interest your paper. It is a well written article including interesting information about a preclinical model for HS. Some points need to be clarified: 

1) What type of surgical procedure was performed in UNTREATED HS patients? It is not common that patients without proper medical treatment to undergo surgery. 

Patients with involvement of several skin locations may exhibit different Hurley stage and IHS4 severity. All selected patients have been under former treatment with clindamycin/rifampicin, which was discontinued before surgery for at last 7 days.

2) Moreover, how perilesional skin was obtained at a distance higher than 5cm? Was it a biopsy or part of the excised surgical piece?

Perlesional skin is obtained at a distance of >5 cm from the visible lesion by a small spindle excision, since punch biopsies would not provide enough tissue to perform these experiments.

3) The specific area of the body from which the skin samples were taken should be included in the article (armpits, groin, other...).

Due to the strict German data protection regulations, patient personal data have to be anonymised when surgical material is provided to the laboratory. Therefore, we cannot recall the patient names in order to extract the surgical locations.

4) References are not well written. Please rewrite them.

We do not understand what the reviewer means with this comment. In any case we checked the references again and made a revision, where we considered it as required.